# Evaluation of the HemoCue® WBC System as a Point of Care Diagnostic Tool for White Blood Cell Quantification in Pinnipeds

Abby M. McClain [1,*], Emily R. Whitmer [2], Carlos Rios [2], Eric D. Jensen [3], Nicole I. Stacy [4] and Shawn P. Johnson [2]

1   National Marine Mammal Foundation, 2240 Shelter Island Drive, Suite 200, San Diego, CA 92106, USA
2   The Marine Mammal Center, 2000 Bunker Road, Sausalito, CA 94965, USA; emily.whitmer@gmail.com (E.R.W.); riosc@tmmc.org (C.R.); shawn@seachangehealth.org (S.P.J.)
3   U.S. Navy Marine Mammal Program, Naval Information Warfare Center Pacific, San Diego, CA 92152, USA; jensene@spawar.navy.mil
4   Department of Comparative, Diagnostic and Population Medicine, College of Veterinary Medicine, University of Florida, Gainesville, FL 32610, USA; stacyn@ufl.edu
*   Correspondence: abby.mcclain@nmmf.org

**Abstract:** Point of care (POC) hematology testing can be valuable in triage and field settings. We assessed the accuracy between the HemoCue® WBC system and two comparative analyzers, as well as the precision of the HemoCue® WBC system in five different pinniped species: *Zalophus californianus*, *Arctocephalus townsendi*, *Callorhinus urcinus*, *Phoca vitulina*, and *Mirounga angustirostris* for white blood cell (WBC) quantification. In *Zalophus* ($n$ = 164; 106 from U.S. Navy Marine Mammal Program (Navy); 58 from The Marine Mammal Center (TMMC)), the HemoCue® was compared to two hematology analyzers, Sysmex Xe-5000 and Vet ABC Plus. In *Phoca* ($n$ = 50; TMMC), *Callorhinus* ($n$ = 29; TMMC), *Arctocephalus* ($n$ = 17; TMMC), and *Mirounga* ($n$ = 67; TMMC), the HemoCue® was compared to Vet ABC Plus only. Bland–Altman and Passing–Bablok agreement of HemoCue® with Sysmex Xe-5000 and Vet ABC Plus analyzers were good for *Zalophus*, *Arctocephalus*, *Phoca*, and *Mirounga* but marginal with *Callorhinus*; bias = $0.56 \times 10^9$/L (*Zalophus*; Navy), $-2.13 \times 10^9$/L (*Zalophus*; TMMC), $-1.59 \times 10^9$/L (*Arctocephalus*), $-2.48 \times 10^9$/L 0.917 (*Phoca*), $-0.01 \times 10^9$/L (*Mirounga*), and $-6.05 \times 10^9$/L (*Callorhinus*). The coefficient of variation from triplicate runs of samples were within acceptable limits for all species ($2.50\% \pm 1.63$ (*Zalophus*; TMMC), $3.09\% \pm 2.14$ (*Arctocephalus*), $2.47\% \pm 1.35$ (*Callorhinus*), $2.88\% \pm 1.75$ (*Phoca*), and $3.44\% \pm 2.53$ (*Mirounga*)), respectively. The presence of nucleated red blood cells (nRBC; 1–37 nRBC/100 WBC) did not significantly interfere with WBC counts in *Zalophus*, *Callorhinus*, and *Phoca* at the population level, but their presence should be evaluated at the individual level. The HemoCue® provides an accurate method for WBC quantification with WBC counts up to $30 \times 10^9$/L (upper limit of linearity of the analyzer) in *Zalophus*, *Arctocephalus*, *Phoca*, and *Mirounga*, but is less accurate in *Callorhinus*, and showed good precision in all species.

**Keywords:** *Arctocephalus*; *Callorhinus*; hematology; *Mirounga*; *Phoca*; point of care; *Zalophus*

## 1. Introduction

Point of care (POC) analyzers offer rapid adjunct diagnostic testing to supplement standard hematology in both human and veterinary medicine. White blood cell (WBC) quantification is a valuable diagnostic marker for the assessment of inflammation and leukogram responses, and the use of POC WBC count analyzers has been implemented in human out-patient and critical care departments, facilitating patient flow, decreasing the length of stay in the emergency department, and aiding in triage [1–3]. Compared to commercial laboratory analyzers in human laboratory diagnostics, POC WBC count

analyzers are cost-effective in resource-limited settings, and provide rapid, bed-side results allowing fast initiation of treatment [1].

In veterinary medicine, the majority of the information regarding POC tests revolves around biochemical analytes (e.g., glucose and lactate) and blood gas analysis rather than hematology [1,4,5]. There are many potential applications for POC WBC count analyzers in domestic and non-domestic species. In a managed care setting and particularly for patients requiring sedation or anesthesia for diagnostics, treatments, and handling, the ability to determine WBC counts patient-side facilitates rapid initiation of treatment, potentially requiring fewer sedation and/or anesthetic events, and can help guide clinical decision-making and management of critically ill patients. In field settings, many health assessment studies in free-ranging wildlife species aim to determine the impact of different diseases on populations but are limited by the lack of capacity to perform a WBC count to complement a WBC differential and often must rely on blood film evaluation only [6,7]. Additionally, a POC WBC count analyzer can provide individual patient-side automated hematology in rescue situations or managed care that may otherwise be impossible.

The HemoCue® WBC system (HemoCue America, Brea, CA 92821, USA) is a lightweight, portable analyzer that is battery powered or can be plugged into an AC 6-volt adapter [8]. The HemoCue® requires a small amount of whole blood (approximately 10 μL), uses a hemolyzing agent to lyse the red blood cells, and a staining agent to color the white blood cells. A photomicroscope is used to capture an image of the stained WBC and the number of cells is counted by image analysis. The WBC count is then displayed on the liquid crystal display screen after approximately three minutes [8]. The presence of platelets, platelet clumps, lipemia, reticulocytes, and immature leukocytes did not interfere with HemoCue® results in humans [8]. However, samples with >2% nucleated red blood cells (nRBC) may give a falsely elevated WBC count [8]. The analyzer should be stored at temperatures between 0 and 50 °C (32–122 °F) with less than 90% non-condensing humidity and operated in temperatures between 15 and 35 °C (59–95 °F) and less than 90% non-condensing humidity [8]. The microcuvettes must be stored at temperatures between 15 and 35 °C (59–95 °F) and less than 90% non-condensing humidity [8]. The system requires very little technical skill and minimal specialized equipment to operate.

Validation studies of the HemoCue® WBC and HemoCue® WBC DIFF differential portable analyzers in humans have demonstrated accuracy for rapid diagnosis of abnormal WBC [3,9,10]. A single study in veterinary medicine showed promising results for the HemoCue® WBC system as a rapid POC test to diagnose abnormal WBC concentrations in cats, horses, cows, and dogs [11]. This study aimed to assess the accuracy (or closeness of agreement) between the HemoCue® WBC system and two comparative analyzers, as well as the precision of the HemoCue® WBC system in five different pinniped species: California sea lions (*Zalophus californianus*), Guadalupe fur seals (*Arctocephalus townsendi*), Northern fur seals (*Callorhinus urcinus*), Pacific harbor seals (*Phoca vitulina*), and Northern elephant seals (*Mirounga angustirostris*).

## 2. Materials and Methods

### 2.1. Animals

Whole blood samples were collected from animals in two separate locations, the U.S. Navy Marine Mammal Program (Navy) and The Marine Mammal Center (TMMC). The Navy houses and cares for a population of *Zalophus* in San Diego Bay (CA, USA). The Marine Mammal Center is a rescue and rehabilitation facility for marine mammals in Sausalito (CA, USA).

#### 2.1.1. U.S. Navy Marine Mammal Program

*Zalophus* samples were the only samples collected from the Navy and included a total of a 106 blood samples from 28 individual *Zalophus* ranging in age from 4 to 32 years old.

### 2.1.2. The Marine Mammal Center

Samples from stranded free-ranging *Zalophus*, *Arctocephalus*, *Callorhinus*, *Phoca*, and *Mirounga* were collected and analyzed at TMMC and age classes were determined based on dentition and body morphometrics in all species [12–14]. Total sample numbers and animals include 58 blood samples from 38 individual *Zalophus* ranging in age class from pup to adult, 17 blood samples from 10 individual *Arctocephalus*, all of which were pups, 29 blood samples from 16 individual *Callorhinus*, all of which were pups, 50 blood samples from 25 individual *Phoca*, all of which were pups, and 66 blood samples from 39 different *Mirounga* ranging in age class from pup to yearling.

### 2.2. Sample Collection, Transport, and Storage

### 2.2.1. U.S. Navy Marine Mammal Program

Samples collected from Navy *Zalophus* were collected opportunistically from an overall clinically healthy population of *Zalophus*, during routine care under the approved U.S. Code, Title 10, USC 7524. Blood was obtained either from the caudal gluteal vein directly into Becton Dickinson (BD) vacutainer tubes containing potassium-ethylenediaminetetraacetic acid (K-EDTA; BD, Franklin Lakes, NJ, USA) using a 20-gauge $\frac{1}{2}$ inch BD vacutainer needle attached to a vacutainer holder or a 21-gauge 2 inch needle attached to a BD vacutainer Luer adapter and vacutainer holder (BD, Franklin Lakes, NJ, USA); or from the superficial antebrachial vein using a 21-gauge $\frac{3}{4}$ inch butterfly needle and syringe, then placed immediately into K-EDTA vacutainer tubes. Gentle inversion of the filled vacutainer tubes was performed for a minimum of one minute prior to analysis for both methods.

### 2.2.2. The Marine Mammal Center

Samples were collected from free-ranging stranded *Zalophus*, *Arctocephalus*, *Callorhinus*, *Phoca*, and *Mirounga* admitted for rehabilitation at TMMC during routine admission, follow-up, or release examinations under the Stranding Agreement between the National Oceanic and Atmospheric Administration's National Marine Fisheries Service West Coast Region and The Marine Mammal Center. Blood was collected from *Zalophus*, *Arctocephalus*, and *Callorhinus* via the caudal gluteal or subclavian vein directly into K-EDTA vacutainer tubes (Oakville, ON LH6 6R5, Canada) using a vacutainer adapter set-up as described above. Blood was collected from *Phoca* and *Mirounga* via the epidural intervertebral sinus directly into K-EDTA vacutainer tubes (Oakville, ON, Canada) using the same vacutainer adapter set-up. Gentle inversion of the filled vacutainer tubes was performed for a minimum of one minute immediately prior to analysis for both methods.

### 2.3. Instrumentation and Analysis

### 2.3.1. U.S. Navy Marine Mammal Program

Blood analysis was performed on well-mixed whole blood in K-EDTA via the Sysmex Xe-5000 (Sysmex Nordic APS Filial Sverige, Marios Gata 13, S-434 37 Kungsbacka, Sweden) comparative instrument at the Naval Medical Center San Diego in single replicate within five hours of collection. White blood cell counts were performed on whole blood samples in K-EDTA by the HemoCue® WBC system candidate instrument in single replicate within one hour of collection. The time span between analyses on the Sysmex Xe-5000 and the HemoCue® ranged from two to four hours for each sample. Blood films prepared from each sample were reviewed to assess the presence of nRBC, red blood cell morphology, and WBC differential and morphology. The WBC counts from both the Sysmex Xe-5000 analyzer and the HemoCue® WBC system were corrected for nRBC when present. Briefly, the HemoCue® analysis process included placing a single drop of whole blood on a sterile flat surface and was taken up by capillary action into the HemoCue® microcuvette. Excess blood was gently wiped away from the cuvette surface prior to being placed into the HemoCue® analyzer. Results were read immediately and recorded.

### 2.3.2. The Marine Mammal Center

Blood analysis was performed on well-mixed whole blood in K-EDTA by the Vet ABC Plus comparative analyzer (Vet ABC Plus; SCIL Vet America, Gurnee, IL 60031, USA) on the dog setting in single replicate within two hours of collection [15,16]. White blood cell counts were performed on whole blood in K-EDTA by the HemoCue® WBC system candidate instrument in triple replicate for each species as described above within two hours of collection. The time span between analyses on the Vet ABC Plus and HemoCue® was less than one hour for each sample. Blood films from each blood sample were reviewed to assess the presence of nRBC, red blood cell morphology, and WBC differential and morphology. White blood cell counts from the Vet ABC Plus analyzer and the HemoCue® WBC system were corrected for nRBC when present.

### 2.4. Reporting and Statistical Analysis

All analyses were performed using R version 3.6.1 [17] and R Studio v 1.2.5019 [18] statistical analysis software. The HemoCue® WBC system corrected WBC count was compared to the comparative instrument's corrected WBC count using Bland–Altman and Passing–Bablok methods and the R package *mcr* [19–21]. Normality of the distribution of the differences between the reference instrument WBC counts and the HemoCue® WBC system WBC counts was confirmed prior to performing analyses. The Bland–Altman plot bias was calculated as the average difference between WBC counts produced by the comparative hematology analyzers and by the HemoCue® WBC system. The Bland–Altman plot limits of agreement were calculated using the equation 1.96 x standard deviation of the average difference between each comparative method and HemoCue® WBC system results in each species [19,22]. To rule out constant and proportional differences between the two methods, the Passing–Bablok intercept and slope 95% confidence intervals must include zero and one, respectively [20]. When interpreting Pearson's correlation coefficient *r*, a very strong association is considered between 0.8 and 0.9, a moderate association is considered between 0.6 and 0.7, a fair association is considered between 0.3 and 0.5, and a poor association is considered between 0.1 and 0.2 [23]. Of note, the upper limit of linearity of the HemoCue® is $30 \times 10^9$/L and a WBC count greater than or equal to $30 \times 10^9$/L will yield a HemoCue® result of "HHH" on the screen. Five blood samples from *Mirounga* yielded results of "HHH" on the HemoCue® and all five samples had a WBC count greater than or equal to $30 \times 10^9$/L via the Vet ABC Plus (WBC counts = 53.4, 39.8, 33.2, 30.2, and $80.0 \times 10^9$/L) comparative instrument. These five samples were removed from the dataset. The lower limit of linearity of the HemoCue® is $0.3 \times 10^9$/L in which a WBC count less than or equal to $0.3 \times 10^9$/L will yield a HemoCue® result of "LLL" on the screen. There were no blood samples in this analysis that yielded results of "LLL".

After the first Bland–Altman and Passing–Bablok analyses were performed on all species, samples that contained nRBC were removed from the dataset and the Bland–Altman and Passing–Bablok analyses were performed again. To assess possible interference caused by nRBC, a Welch's two sample t-test was performed to compare significant differences in the biases. Nucleated red blood cells were present in samples from two *Phoca* and four *Callorhinus* at The Marine Mammal Center and eight *Zalophus* at the Navy.

The precision of the HemoCue® WBC system was assessed for all five species at The Marine Mammal Center using triplicate tests performed on the same blood sample. The coefficient of variation was calculated for each triplicate group by dividing the standard deviation of the three measurements by the mean of the three measurements. The average coefficient of variation, standard deviation, and range were calculated. The coefficient of variation can be expressed as a percentage where <10% is considered to be very good, 10–20% is considered to be good, 20–30% is acceptable, and >30% is unacceptable [24]. In addition, the total observed error (TEobs) for the WBC count was calculated for all five species in this study using the equation:

$$TEobs = absolute\ bias\% + 2CV$$

and compared to the total allowable error (TEa) recommended by guidelines set forth by the American Society for Veterinary Clinical Pathology (ASVCP) [25].

## 3. Results

Overall, the HemoCue® performed well over a large range of WBC counts ($2.5-30 \times 10^9$/L) in four out of five different species of pinnipeds (Table 1).

### 3.1. Zalophus—Navy

Both the Bland–Altman analysis and Passing–Bablok regression analysis showed strong correlations between the comparative instrument and the HemoCue® WBC system (Table 1). The bias determined from the Bland–Altman plot was $0.56 \times 10^9$/L. Of the 106 blood samples analyzed, seven data points fell outside of the upper and lower limits of agreement (Figure 1A). There was no significant proportional bias between the two methods; however, a constant difference was observed. The HemoCue® overall had WBC counts slightly lower than the Sysmex Xe-5000 WBC count (Figure 2A). Nine samples with nRBC ranging from 1 to 3 nRBC/100 WBC were identified by blood film evaluation. The WBC count from both the HemoCue® WBC system and the Sysmex Xe-5000 were corrected for nRBC prior to performing the Bland–Altman and Passing–Bablok analyses. Additionally, the samples with nRBC were analyzed on an individual basis. The differences between the Sysmex Xe-5000 WBC counts and the HemoCue® WBC counts for the nine samples that contained nRBC prior to adjusting the HemoCue® WBC counts were: 0.7, 0.7, 0.5, 0.4, 0.8, 0.3, 0.9, 1.1, and $1.2 \times 10^9$/L. After adjusting the HemoCue® WBC counts for the presence of nRBC, the differences between the Sysmex Xe-5000 and the HemoCue® WBC counts were approximately the same: 0.7, 0.7, 0.5, 0.4, 0.8, 0.4, 0.9, 1.1, and $1.2 \times 10^9$/L, respectively. The Navy samples were analyzed in single runs, so a coefficient of variation was not able to be performed at this location.

### 3.2. Zalophus—The Marine Mammal Center

A strong correlation between the Vet ABC Plus analyzer and the HemoCue® was observed (Table 1). The bias determined from the Bland–Altman plot was $-2.13 \times 10^9$/L. Of the 58 blood samples analyzed, two data points fell outside of the upper and lower limits of agreement (Figure 1B). There was no significant constant difference between the two methods; however, a proportional bias was observed. As the magnitude of WBC counts increased, the difference between the Vet ABC Plus analyzer and the HemoCue® increased (Figure 2B). Nucleated RBC were absent in all samples from this group.

All 58 *Zalophus* blood samples were run in triplicate to assess the precision of the HemoCue®. The HemoCue® was precise in *Zalophus* with a coefficient of variation of 2.5%. The TEobs for *Zalophus* was 13.66%.

### 3.3. Arctocephalus—The Marine Mammal Center

High agreement between the HemoCue® and the Vet ABC Plus was observed in *Arctocephalus* (Table 1). The bias determined from the Bland–Altman plot was $-1.59 \times 10^9$/L. All 17 blood sample data points were within the upper and lower limits of agreement (Figure 1C). There was no significant constant difference between the two methods; however, a proportional bias was observed. Similar to what was observed in *Zalophus*, as the magnitude of WBC counts increased, the difference between the Vet ABC Plus and the HemoCue® increased (Figure 2C). The sample size for *Arctocephalus* was the smallest of all five species and likely contributed to the wide CI observed for the Passing–Bablok intercept. Nucleated RBC were absent in all samples from this group.

Five *Arctocephalus* blood samples were run in triplicate to assess the precision of the HemoCue®. The HemoCue® was precise in *Arctocephalus* with a coefficient of variation of 3.09% and the TEobs for *Arctocephalus* was 18.48%.

**Table 1.** Summary of method comparison results for white blood cell counts in five species of pinnipeds: California sea lions (*Zalophus californianus*) from the U.S. Navy Marine Mammal Program (CSL MMP), California sea lions from The Marine Mammal Center (CSL TMMC), Guadalupe fur seals (*Arctocephalus townsendi*; GFS; TMMC), Northern fur seals (*Callorhinus ursinus*; NFS; TMMC), harbor seals (*Phoca vitulina*; HS; TMMC), and Northern elephant seals (*Mirounga angustirostris*; ES; TMMC).

| Species (Location) | Number of Samples (Individual Animals) | Bland–Altman | | | Passing–Blok | | | Precision | nRBC |
| | | Bias (SD) [a] | Upper LOA [c] | Lower LOA [c] | Slope (CI) [b] | Intercept (CI) [b] | Pearson's r | Coefficient of Variation (SD) [a] | Range of nRBC/100 WBC (n) |
|---|---|---|---|---|---|---|---|---|---|
| *Zalophus* (Navy) | 106 (28) | 0.56 (0.62) | 1.77 | −0.64 | 1.00 (0.96, 1.08) | −0.65 (−1.04, −0.44) | 0.969 | NA | 1–3 (9) |
| *Zalophus* (TMMC) [d] | 58 (38) | −2.13 (1.96) | 1.72 | −5.98 | 1.22 (1.08, 1.40) | −0.42 (−2.00, 0.65) | 0.941 | 2.50 (1.63) | NA |
| *Arctocephalus* (TMMC) [d] | 17 (10) | −1.59 (1.26) | 0.88 | −4.06 | 1.51 (1.12, 2.72) | −0.99 (−6.69, 1.06) | 0.968 | 3.09 (2.14) | NA |
| *Callorhinus* (TMMC) [d] | 29 (16) | −5.67 (4.13) | 2.43 | −13.76 | 1.40 (1.07, 1.72) | 0.35 (−3.71, 5.28) | 0.702 | 2.47 (1.35) | 5–37 (4) |
| *Phoca* (TMMC) [d] | 50 (25) | −2.34 (2.49) | 2.55 | −7.23 | 1.27 (1.19, 1.35) | −0.77 (−1.42, −0.10) | 0.912 | 2.88 (1.75) | 8–28 (2) |
| *Mirounga* (TMMC) [d] | 67 (40) | −0.01 (2.62) | 5.13 | −5.16 | 0.97 (0.89, 1.05) | 0.82 (0.03, 1.81) | 0.936 | 3.44 (2.53) | NA |

([a]) standard deviation; ([b]) confidence interval; ([c]) limit of agreement; ([d]) The Marine Mammal Center. Units: Bias, standard deviation, and confidence interval (cells $\times 10^9$/L); limits of agreement (cells $\times 10^9$/L); coefficient of variation (%).

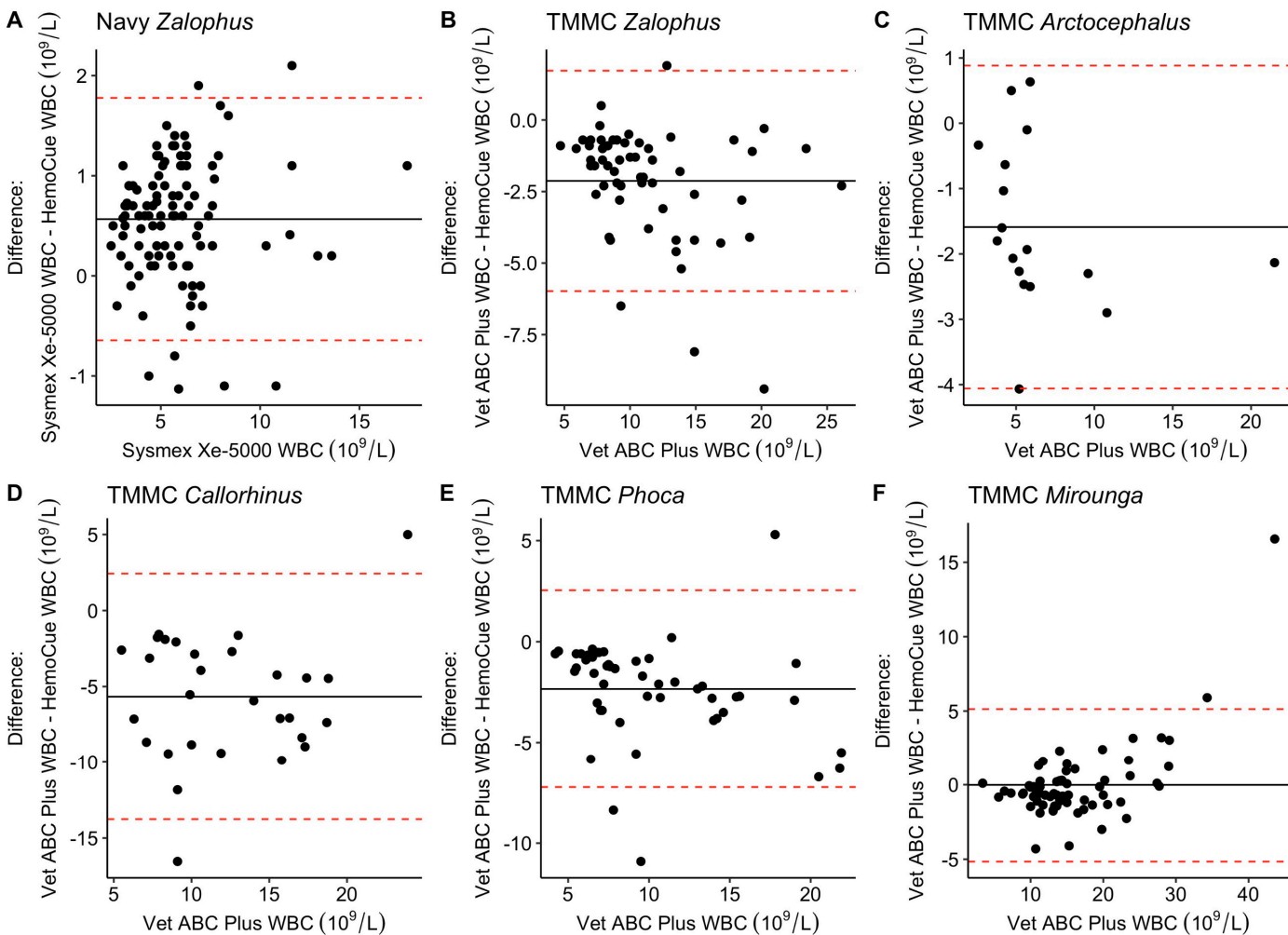

**Figure 1.** Bland-Altman plots comparing the corrected WBC counts between the comparative analyzers and the HemoCue® WBC system for each species. (**A**) Bland-Altman plot of U.S. Navy Marine Mammal Program California sea lion (*Zalophus californianus*) samples comparing the Sysmex Xe-5000 automated hematology analyzer to the HemoCue® WBC system. The Sysmex Xe-5000 WBC count is depicted on the x-axis. The difference between the Sysmex Xe-5000 WBC count and the HemoCue® WBC system WBC count is depicted on the y-axis. The solid black line indicates the overall bias, and the red dashed lines indicate the upper and lower limits of agreement. (**B**) Bland-Altman plot of The Marine Mammal Center (TMMC) *Zalophus* samples comparing the Vet ABC Plus automated hematology analyzer to the HemoCue® WBC system. The Vet ABC Plus WBC count is depicted on the x-axis. The difference between the Vet ABC Plus WBC count and the HemoCue® WBC system WBC count is depicted on the y-axis. (**C**) Bland-Altman plot of TMMC Guadalupe fur seal (*Arctocephalus townsendi*) samples comparing the Vet ABC Plus automated hematology analyzer to the HemoCue® WBC system. (**D**) Bland-Altman plot of TMMC Northern fur seal (*Callorhinus ursinus*) samples comparing the Vet ABC Plus automated hematology analyzer to the HemoCue® WBC system. (**E**) Bland-Altman plot of TMMC Harbor seal (*Phoca vitulina*) samples comparing the Vet ABC Plus automated hematology analyzer to the HemoCue® WBC system. (**F**) Bland-Altman plot of TMMC Northern elephant seal (*Mirounga angustirostris*) samples comparing the Vet ABC Plus automated hematology analyzer to the HemoCue® WBC system.

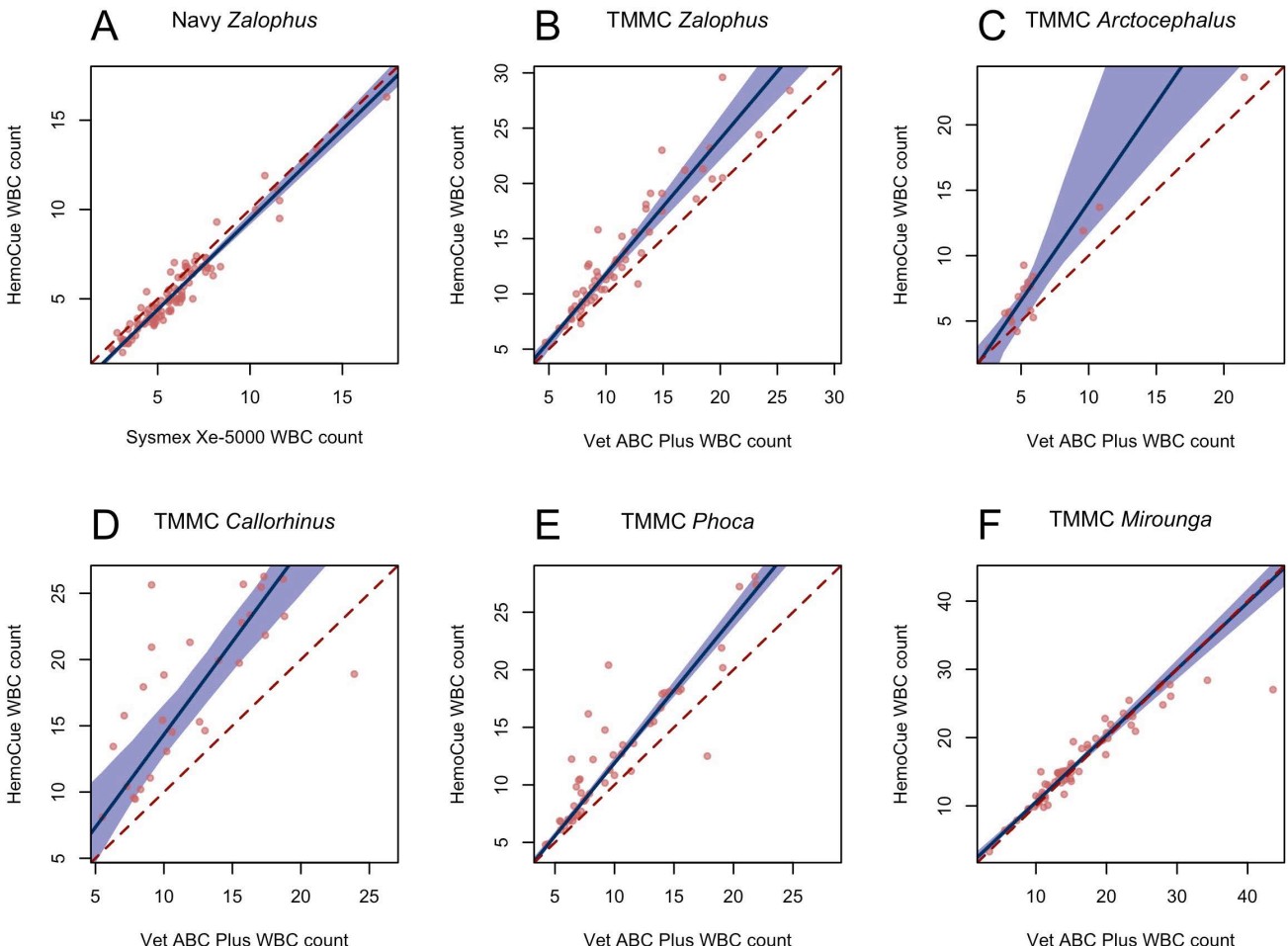

**Figure 2.** Passing-Bablok plots comparing the corrected WBC counts between the comparative analyzers and the HemoCue® WBC system for each species. Units for both comparative instruments and the HemoCue® WBC System are ($10^9$/L). (**A**) Passing-Bablok regression of the U.S. Navy Marine Mammal Program California sea lion (*Zalophus californianus*) samples comparing the Sysmex Xe-5000 to the HemoCue® WBC system. The WBC count from the Sysmex Xe-5000 is depicted on the x-axis. (**B**) Passing-Bablok regression of The Marine Mammal Center (TMMC) *Zalophus* samples comparing the Vet ABC Plus automated hematology analyzer to the HemoCue® WBC system. The WBC counts from the Vet ABC Plus are depicted on the x-axis. The WBC counts from the HemoCue® WBC system are depicted on the y-axis. (**C**) Passing-Bablok regression of TMMC Guadalupe fur seal (*Arctocephalus townsendi*) samples comparing the Vet ABC Plus automated hematology analyzer to the HemoCue® WBC system. The WBC counts from the Vet ABC Plus are depicted on the x-axis. The WBC counts from the HemoCue® WBC system are depicted on the y-axis. (**D**) Passing-Bablok regression of TMMC Northern fur seal (*Callorhinus ursinus*) samples comparing the Vet ABC Plus automated hematology analyzer to the HemoCue® WBC system. The WBC counts from the Vet ABC Plus are depicted on the x-axis. The WBC counts from the HemoCue® WBC system are depicted on the y-axis. (**E**) Passing-Bablok regression of TMMC harbor seal (*Phoca vitulina*) samples comparing the Vet ABC Plus automated hematology analyzer to the HemoCue® WBC system. The WBC counts from the Vet ABC Plus are depicted on the x-axis. The WBC counts from the HemoCue® WBC system are depicted on the y-axis. (**F**) Passing-Bablok regression of TMMC.

### 3.4. Callorhinus—The Marine Mammal Center

Marginal agreement between the HemoCue® and the Vet ABC Plus was observed in *Callorhinus* (Table 1). The bias determined from the Bland–Altman plot was $-5.67 \times 10^9$/L. Of the 29 blood samples, two data points fell outside of the upper and lower limits of

agreement (Figure 1D). There was no significant constant difference between the two methods; however, a proportional difference was observed. Similar to what was observed in *Zalophus* and *Arctocephalus*, as the magnitude of WBC counts increased, the difference between the Vet ABC Plus and the HemoCue® increased (Figure 2D). Four samples with nRBC ranging from 5 to 37 nRBC/100 WBC were identified on blood film. The WBC count from both the HemoCue® WBC system and the Vet ABC Plus were corrected for nRBC prior to performing the Bland–Altman and Passing–Bablok analyses. Additionally, the samples with nRBC were analyzed on an individual basis. The differences between the Vet ABC Plus WBC counts and the HemoCue® WBC counts for the four samples that contained nRBC prior to adjusting the HemoCue® WBC counts were: 2.0, 10.4, 11.1, and $10.3 \times 10^9$/L. After adjusting the HemoCue® WBC counts for the presence of nRBC, the differences between the Vet ABC Plus and the HemoCue® WBC counts were slightly improved, but overall, still relatively large in magnitude at: 4.9, 8.3, 9.8, and $9.4 \times 10^9$/L, respectively.

Twenty-eight *Callorhinus* blood samples were run in triplicate to assess the precision of the HemoCue®. The HemoCue® was precise in *Callorhinus* with a coefficient of variation of 2.47%. Total observed error for *Callorhinus* was 41.40%.

### 3.5. Phoca—The Marine Mammal Center

High agreement between the HemoCue® and the Vet ABC Plus was observed in *Phoca* (Table 1). The bias determined from the Bland–Altman plot was $-2.34 \times 10^9$/L. Of the 50 blood samples, three data points fell outside of the upper and lower limits of agreement (Figure 1E). There were both constant and proportional differences between the two testing methods in *Phoca*. Similar to what was observed in *Zalophus*, *Arctocephalus*, and *Callorhinus*, as the magnitude of WBC counts increased, the difference between the Vet ABC Plus and the HemoCue® increased (Figure 2E). Two samples with nRBC, 8 and 28 nRBC/100 WBC, were identified on blood film. The WBC count from both the HemoCue® WBC system and the Vet ABC Plus were corrected for nRBC prior to performing the Bland–Altman and Passing–Bablok analyses. Additionally, the samples with nRBC were analyzed on an individual basis. The differences between the Vet ABC Plus WBC counts and the HemoCue® WBC counts for the two samples that contained nRBC prior to adjusting the HemoCue® WBC counts were 5.7 and $8.9 \times 10^9$/L. After adjusting the HemoCue® WBC counts for the presence of nRBC, the differences between the Vet ABC Plus and the HemoCue® WBC counts were improved at 1.1 and $6.7 \times 10^9$/L, respectively.

Thirty-two *Phoca* blood samples were run in triplicate to assess the precision of the HemoCue®. The HemoCue® was precise in *Phoca* with a coefficient of variation of 2.88% and TEobs for *Phoca* was 17.35%.

### 3.6. Mirounga—The Marine Mammal Center

High agreement between the HemoCue® and the Vet ABC Plus was observed in *Mirounga* (Table 1). The bias determined from the Bland–Altman plot was $-0.01 \times 10^9$/L. Of the 66 blood samples analyzed, two data points fell outside of the upper and lower limits of agreement (Figure 1F). In *Mirounga*, there was a constant but no proportional bias between the Vet ABC Plus and the HemoCue® (Figure 2F) and likely due to a single outlier in which the difference between the comparative instrument and the HemoCue® was $16.56 \times 10^9$/L. Nucleated RBC were absent in all samples from this group.

Fifty-six *Mirounga* blood samples were run in triplicate to assess the precision of the HemoCue® for *Mirounga* samples. The HemoCue® was precise in *Mirounga* with a coefficient of variation of 3.44%. The TEobs for *Mirounga* was 6.79%.

## 4. Discussion

This study demonstrates the accuracy and precision of the HemoCue® WBC system in five species of pinnipeds: *Zalophus*, *Arctocephalus*, *Callorhinus*, *Phoca*, and *Mirounga* over a wide range of WBC counts up to $30 \times 10^9$/L in comparison to two comparative instruments (Vet ABC Plus and Sysmex Xe-5000). Based on the results of this study, this

portable and easy-to-use POC WBC quantification device offers an additional diagnostic tool and potential applications to zoological and wildlife medicine, including but not limited to health assessments in the field, patient-side diagnosis of abnormal WBC counts leading to more rapid and guided treatment decisions, and in clinical facilities limited by financial resources.

When compared to the Vet ABC Plus, the HemoCue® had marginal to good accuracy in all five species with the best accuracy observed in *Mirounga* and the lowest accuracy observed in *Callorhinus*. Very good accuracy was observed in *Arctocephalus*; however, a very wide confidence interval in the Passing–Bablok regression was observed, likely due to the small sample size. When compared to the Sysmex Xe-5000 in *Zalophus* only, the HemoCue® also demonstrated very good accuracy. The variability observed between the two comparative analyzers and the HemoCue® in this study was similar to what was observed by Riond et al. (2012), where WBC counts for feline, equine, and bovine species were more accurate than for dogs which showed marginal accuracy [11]. In this study, the HemoCue® demonstrated very good precision in all five species of pinnipeds.

It is important to note that the HemoCue® WBC system, despite showing good accuracy, did show constant and/or proportional differences in comparison to both comparative instruments in all five species in this study. On average, the HemoCue® tended to underestimate WBC concentrations compared to the Sysmex Xe-5000 and overestimate WBC counts compared to the Vet ABC Plus. Results observed in rehabilitating *Zalophus* at TMMC in which overall WBC counts were higher compared to the overall healthy Navy *Zalophus*, showed that as the magnitude of WBC counts increased, the HemoCue® became less accurate when compared to the Vet ABC Plus. Results from *Arctocephalus* and *Callorhinus* at TMMC showed similar results to *Zalophus* in which an increase in magnitude of WBC counts corresponded to less accurate HemoCue® results as compared to the Vet ABC Plus. Results from *Phoca* and *Mirounga* at TMMC showed a constant difference. In *Phoca*, the HemoCue® on average estimated the WBC count $2.5 \times 10^9$/L greater than the results of the Vet ABC Plus. However, the magnitude of WBC counts did not affect the accuracy of the HemoCue® in *Phoca* as it did in *Zalophus*, *Arctocephalus*, and *Callorhinus*. In *Mirounga*, the constant bias was likely due to a single outlier in which the difference between the Vet ABC Plus analyzer and the HemoCue® was $16.56 \times 10^9$/L, a difference that was three times greater than any other analyzed sample and from a patient with a markedly elevated WBC count of $43.6 \times 10^9$/L. The ASVCP Quality Assurance and Laboratory Standards (QALS) accepts TEa for WBC counts at a reference laboratory of 15% and at an in-clinic laboratory of 20% [25]. For *Zalophus*, *Arctocephalus*, *Phoca*, and *Mirounga*, the TEobs was less than the TEa for in-clinic laboratories (13.66%, 18.48%, 17.35%, and 6.79%, respectively), complementing observations by Bland–Altman plots and Passing–Bablok analyses. However, for *Callorhinus*, TEobs (41.40%) was greater than the recommended TEa for in-clinic laboratories, complementing what was observed in the Bland–Altman plots and Passing–Bablok analysis and indicating the HemoCue® requires more testing before it can confidently be used as a POC WBC device for *Callorhinus*. The reason for the decreased HemoCue® accuracy observed in this study in *Callorhinus* is unknown but may be due in part to the small sample size analyzed. The TEobs was unable to be calculated for *Zalophus* in comparison to the Sysmex Xe-5000 as the HemoCue® samples were performed in single runs and a coefficient of variation was unable to be calculated.

A study performed in humans found that the presence of more than 2% circulating nRBC can cause inaccurate HemoCue® results [3]. However, in the current study at the population level, the presence of nRBC (maximum nRBC/100 WBC = 37) did not change the observed constant or proportional biases. Overall, the sample sizes for each species were relatively small with few animals having nRBC present in each dataset. The small number of animals with nRBC may have been the reason why nRBC did not have a significant effect on the HemoCue® accuracy in this study rather than a true lack of interference. At the individual level, after adjusting the HemoCue® WBC counts for the presence of nRBC, the magnitude of the difference between the comparative analyzers and the HemoCue® results

improved, most notably for *Phoca*, but were still relatively large overall. Blood film review to identify nRBC and correct the WBC count should always be performed, especially with unexpected HemoCue® results.

The HemoCue® was demonstrated to be an accurate and precise diagnostic test for WBC quantification in *Zalophus*, *Arctocephalus*, *Phoca*, and *Mirounga*, and showed marginal accuracy but good precision in *Callorhinus*. In all cases, the WBC counts must be clinically interpreted in light of overall patient health status, findings from physical examination, and additional diagnostic tests, including blood film review (e.g., WBC estimate as control measure, white blood cell differential, and evaluation of blood cell morphology). If an unexpected result is obtained by the HemoCue®, we recommend in addition to blood film review that at least one other method of hematological analysis be performed to corroborate the HemoCue®, e.g., repeat HemoCue® test or submission to a reference laboratory. Additionally, confirmation of results is highly recommended for animals that are documented or suspected to have an increased number of nRBC.

Based on the results observed in this study, the HemoCue® WBC system is a promising tool that has the potential to estimate WBC concentrations accurately and precisely in *Zalophus*, *Arctocephalus*, *Phoca*, and *Mirounga*, but more testing is recommended for *Callorhinus*. The HemoCue® WBC system unit costs approximately USD 2000.00 and each cuvette costs approximately USD 2.85. When combined with other analyses that are readily performed at low cost (e.g., packed cell volume, total protein, and manual differential), the HemoCue® allows for a fairly comprehensive, rapid, and low-cost hematological assessment for four out of five pinniped species analyzed in this study.

**Author Contributions:** All authors had significant contributions to this study. Conceptualization, A.M.M., E.R.W., N.I.S., S.P.J., E.D.J.; Methodology, A.M.M., E.R.W., N.I.S.; Software, A.M.M.; Validation, E.R.W., C.R., N.I.S.; Formal analysis, A.M.M.; Investigation, A.M.M., C.R.; Resources, S.P.J., E.D.J.; Data curation, A.M.M., C.R.; Writing—original draft, A.M.M.; Writing—review and editing, E.R.W., C.R., N.I.S., S.P.J., E.D.J.; Visualization, A.M.M., N.I.S.; Supervision, N.I.S., S.P.J., E.D.J.; Project administration, A.M.M., E.R.W., C.R. All authors have read and agreed to the published version of the manuscript.

**Funding:** This research received no external funding.

**Acknowledgments:** The authors wish to sincerely thank the U.S. Navy Marine Mammal Program and The Marine Mammal Center's dedicated veterinary staff, trainers, and volunteers for providing the highest standard of care for both managed and stranded pinnipeds. This is scientific contribution number 328 of the National Marine Mammal Foundation. Graphical abstract created using biorender. com.

**Conflicts of Interest:** The authors declare no conflict of interest.

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
