# Peer review of "Evaluation of the HemoCue® WBC System as a Point of Care Diagnostic Tool for White Blood Cell Quantification in Pinnipeds"

_2673-1924, doi:10.3390/oceans3010007_

Round 1

Reviewer 1 Report

This is an excellent manuscript evaluating a point of care diagnostic tool for white blood cell quantification against routine WBC analyzers. I commend the advanced statistical evaulation. A few minor suggestions:

 - In Table 1 please adjust column width so that with the cell "coefficient of variation"  - coefficient is not split in two lines.

  • It might be good to state in general the age classes ranges of the animals used in the study. I imagine that the California sea lions were of more varied age classes compared to the other species that may be mostly pups or yearlings. It may be important to note that age class didn't affect the results of this comparision.  

Author Response

The authors sincerely thank the reviewer for taking the time to review and recommend revisions for their manuscript.

Reviewer #1:

- In Table 1 please adjust column width so that with the cell “coefficient of variation” - coefficient is not split in two lines

The authors appreciate the reviewer’s attention to detail and recommendations. The authors have adjusted the column width so “coefficient” is all on one line now.

- It might be good to state in general the age classes ranges of the animals used in the study. I imagine that the California sea lions were of more varied age classes compared to the other species that may be mostly pups or yearlings. It may be important to note that age class didn’t affect the results of this comparison.

The authors appreciate the reviewer’s recommendation and have added the specific age/age class ranges for each species. In addition, the authors have added in references for determining the age class in free-ranging pinnipeds.

Reviewer 2 Report

The study compares automated WBC count from 5 pinniped species obtained by the HemoCue to two analyzers: the Vet ABC Plus. The study also compares WBC count from one species using the HemoCue and Sysmex. The authors report accuracy and precision.

The manuscript is well written, and for a study involving non-domestic species has relatively large sample sizes.  A point-of-care analyzer to determine WBC count has potential clinical use in a variety of resource-poor conditions. Overall, the methods and interpretations are solid; however several changes are recommended to improve the manuscript.

1) The abstract seems unnecessarily complex, and it is difficult to understand what comparisons were made in each species. Additionally the use of abbreviations for the species is difficult to follow, and not used in the remainder of the manuscript. Unless these are commonly used in pinniped research articles, recommend using genus (as in main body) or more standard (Z. califonianus).

2) Has the VetABC Plus been validated for accuracy/precision in pinnipeds? If so, please cite. If not, please at least cite its validation in other veterinary species. Without this information, it’s difficult to ascertain whether the HemoCue accurately determine WBC count, or just accurately reflects a (potentially erroneous) WBC count obtained on the VetABC Plus.

3) Effect of nRBCs: Authors note in abstract and results that nRBC counts up to 37/100 WBC did not significantly affect WBC counts, which is difficult to believe and not supported by the statistical test used to generate this conclusion. The Welch’s t-test compares two WBC counts between two populations (in this case all HemoCue WBC +/- removal of patients with nRBCs). Given the relatively low number of patients with nRBCs, it is not surprising that there wouldn’t be a statistically significant difference between WBC count at the population level, a point alluded to in the discussion. Since we are more usually more interested in WBC count in an individual, it would be more relevant to examine at the individual level. Why didn’t investigators correct the HemoCue WBC count, similar to what was done for the reference analyzer WBC counts? Notably, the authors recommended doing just this in the discussion (Lines 370-372). This discussion point is not currently supported by the current results, but could be if the data are re-analyzed. I assume that the HemoCue is not able to exclude nRBCs from the WBC count?

4) In veterinary laboratory medicine, “systematic error” includes both proportional bias and constant (or fixed) bias. I’ve rarely seen “systematic” used interchangeably with constant error, but this is not in keeping with standard language, so recommend changing “systematic” to either “constant” of “fixed” throughout manuscript. (Reference: Jensen AL, Kjelgaard-Hansen M. Method comparison in the clinical laboratory. Vet Clin Pathol. 2006 Sep;35(3):276-86. doi: 10.1111/j.1939-165x.2006.tb00131.x. PMID: 16967409.)

5) Throughout results section, the authors refer to Table 1 for evidence of strong correlation, but correlation data are not included in Table 1. Recommend including Pearson’s correlation r in Table 1, rather than in Figure 2. It doesn’t makes sense to include Pearson’s r in the Passing-Bablock plots in Figure 2.

Minor:

Lines 47-55: Any literature to cite for this section? How could POC determination of WBC help in patients that need to be sedated for assessment? Wouldn’t they need to be sedated for blood collection? Any literature to support statement that field researchers need WBC count?

Line 331-333: Replicated words in sentence

Line 375: Do you mean “accuracy” rather than “reliability”?

Author Response

The authors sincerely appreciate and want to thank the reviewer for taking the time to thoroughly review and make recommendations for editing and improving their manuscript.

Reviewer #2

- The abstract seems unnecessarily complex and it is difficult to understand what comparisons were made in each species. Additionally the use of abbreviations for the species is difficult to follow and not used in the remainder of the manuscript. Recommend using genus (as in main body) or more standard (Z. californianus)

The authors appreciate the reviewer’s comments and have adjusted the abstract as described below to aim to make it more easily understood. The authors have changed all abbreviations to genus names as in the main body of the manuscript and have edited the abstract to remove some potentially duplicated information with regards to the genus names stated multiple times throughout the abstract in potentially unnecessary locations. The authors have worked to keep all the relevant information that is needed in the abstract to outline the manuscript as well.

- Has the VetABC Plus been validated for accuracy/precision in pinnipeds? If so, please cite. If not, please at least cite its validation in other veterinary species. Without this information, it’s difficult to ascertain whether the HemoCue accurately determine WBC count, or just accurately reflects a (potentially erroneous) WBC count obtained on the VetABC Plus - use the CRC to find references for ranges

The Vet ABC Plus has been used in multiple exotic species to help develop reference ranges for the species. The authors have included two references where the Vet ABC Plus was used to develop reference ranges for Pacific Harbor seals and California sea lions (Greig et al. 2010; Williams 2013). Additionally, The Marine Mammal Center has developed in-house reference ranges for Northern elephant seals using the Vet ABC Plus, but this data has not been published. The authors have also included which setting the Vet ABC Plus is run on when analyzing the pinniped samples at The Marine Mammal Center.

- Effect of nRBCs: Authors note in abstract and results that nRBC counts up to 37/100 WBC did not significantly affect WBC counts, which is difficult to believe and not supported by the statistical test used to generate this conclusion. The Welch’s t-test compares two WBC counts between two populations (in this case all HemoCue WBC +/- removal of patients with nRBCs). Given the relatively low number of patients with nRBCs, it is not surprising that there wouldn’t be a statistically significant difference between WBC count at the population level, a point alluded to in the discussion. Since we are usually more interested in WBC count in an individual, it would be more relevant to examine at the individual level. Why didn’t investigators correct the HemoCue WBC count, similar to what was done for the reference analyzer WBC counts? Notably, the authors recommend doing just this in the discussion (Lines 370-372). This discussion point is not currently supported by the current results, but could be if the data are re-analyzed. I assume that the HemoCue is not able to exclude nRBCs from the WBC count?

The reviewer is correct in that the HemoCue is not able to exclude nRBCs from the WBC count although the HemoCue reagents do lyse the RBC and dye the WBC prior to counting the cells. The HemoCue operating manual states that >2% nRBC in a sample can affect the results of the HemoCue.

The authors sincerely appreciate the reviewer’s comments and recommendations regarding this topic as it is extremely important for WBC quantification devices. The authors have re-analyzed the data and have corrected the HemoCue WBC counts for the presence of nRBC in each sample they were present regardless of the magnitude of nRBC. The authors have reported the differences between the respective analyzer (Sysmex Xe-5000 and/or the Vet ABC Plus) and the HemoCue WBC counts prior to correcting the WBC count as well as the differences between the respective analyzer and the HemoCue after correcting the HemoCue WBC count for the presence of nRBC in the results section. The authors have currently left the t-test results in the text so the readers can get an idea on the population level as well as on an individual level with the newly reported differences.

- In veterinary laboratory medicine, “systematic error” includes both proportional bias and constant (or fixed) bias. I’ve rarely seen “systematic” used interchangeable with constant error, but this is not in keeping with standard language, so recommend changing “systematic” to either “constant” or “fixed” throughout manuscript.

The authors sincerely appreciate the explanation by the reviewer and have reviewed the recommended reference again. The authors have changed “systematic” to “constant” throughout the manuscript as suggested by the reviewer throughout the text. These changes have been left in track changes so the reviewer can see where these have been made.

- Throughout results section, the authors refer to Table 1 for evidence of strong correlation, but correlation data are not included in Table 1. Recommend including Pearson’s correlation r in Table 1, rather than in Figure 2. It doesn’t make sense to include Pearson’s r in the Passing-Bablok plots in Figure 2.

The authors thank the reviewer for pointing this out and adding an explanation and have added Pearson’s r value as a column under the Passing Bablok main heading in Table 1 per the recommendation of the reviewer. The authors have also removed Pearson’s r value from each of the plots in Figure 2.

Minor revisions:

- Lines 47-55: Any literature to cite for this section? How could POC determination of WBC help in patients that need to be sedated for assessment? Wouldn’t they need to be sedated for blood collection? Any literature to support statement that field researchers need WBC count?

The authors appreciate the reviewer’s questions regarding this topic and have added two references regarding health assessments in free-ranging pinnipeds in which reference ranges were established in the species in question to help determine the effects of disease on the species. In both studies, hematology biomarkers needed to be assessed during the health assessments in which the HemoCue could have been useful. The authors have also added material indicating how a POC WBC device could help decrease the number of sedations/anesthesias needed for patients requiring sedation/anesthesia for handling and treatments. The reviewer is correct in stating that the patient would need to be sedated for the blood draw initially, however with the HemoCue, treatment could be initiated during that one anesthetic event pending the HemoCue results instead of waking the animal up, waiting for haematology results from a comparative analyzer, and having to re-anesthetize the animal to initiate treatment.

- Line 331 - 333: Replicated words in sentence

The authors suspect the reviewer meant the word “bias” that was used three times in the sentence describing what the units were for the table. The authors have removed the word “bias” from in front of “standard deviation” and “confidence intervals” in the table description section and have left these changes in track changes so the reviewer can see them more easily.

- Line 375: Do you mean “accuracy” rather than “reliability”

The authors have edited this word and changed “reliability” to “accuracy” in this sentence and thank the reviewer for their recommendation.

Round 2

Reviewer 2 Report

The authors have strengthened the manuscript, and this reviewer appreciates authors' efforts to conduct the study and write up the findings. The manuscript is close to being ready, but I still have an outstanding question/comment about the analysis with or without/nRBCs. 

It doesn’t appear that the authors re-analyzed the data after correcting HemaCue counts for nRBCs (all numerical data are the same and graphs look the same). Absolute differences are in the text, but all numerical values in Table 1 and Figures  look the same as the first version. It’s not why authors would not assess for bias using the corrected WBC counts from each analyzer (currently comparing corrected Sysmex WBC count to uncorrected HemaCue count). If anything, it is likely that re-analyzing with corrected counts would strengthen correlation and decrease bias. Please indicate rationale for not not comparing corrected WBC counts, and make clear exactly in the legend and text what values (corrected versus uncorrected) comparison is being shown.

Minor point: The sentence with repeated words is still in the manuscript (new lines 358-360): “In this study the HemoCue demonstrated very good precision in all five species of pinnipeds, the HemoCue demonstrated very good precision.”

Author Response

The authors wish to thank the reviewer again for their attention to detail after the first round of revisions and making more recommendations regarding the important topic of the presence of nRBC when assessing hematology analyzers. The authors think the manuscript is improved due to the recommendations of the reviewer. 

  • It doesn’t appear that the authors re-analyzed the data after correcting HemoCue counts for nRBCs (all numerical data are the same and graphs look the same). Absolute differences are in the text, but all numerical values in Table 1 and Figures look the same as the first version. It’s not why authors would not assess for bias using the corrected WBC counts from each analyzer (currently comparing corrected Sysmex WBC count to uncorrected HemoCue count). If anything, it is likely that re-analyzing with corrected counts would strengthen correlation and decrease bias. Please indicate rationale for not comparing corrected WBC counts, and make clear exactly the legend and text what values (corrected versus uncorrected) comparison is being shown.

The authors appreciate the reviewer’s comments and further questions regarding this topic and apologize that they misunderstood the original comment from round 1 of revisions. The authors completely agree that analyzing the data after correcting for the presence of nRBC will improve the manuscript significantly compared to the first version. The authors have re-analyzed the data from the Navy Zalophus, TMMC Callorhinus, and TMMC Phoca and have edited the variables within Table 1 and the text to reflect the changes. The authors have also added a sentence in the description of both Figure 1 and Figure 2 stating the analyses are assessing the corrected WBC counts of the comparative analyzers and the HemoCue. The authors have currently left in the absolute differences for the readers to see what the differences were prior to correcting the HemoCue WBC counts and then after correcting the WBC counts in the results section. The materials and methods have also been adjusted to reflect the analyses being performed after correcting for the presence of nRBC.  

  • Minor point: The sentence with repeated words is still in the manuscript (new lines 358-360)

The authors thank the reviewer for catching this again and have removed the repeated words from the appropriate sentence.
